# Brain–Computer Interface and Hand-Guiding Control in a Human–Robot Collaborative Assembly Task

Yevheniy Dmytriyev , Federico Insero , Marco Carnevale and Hermes Giberti *

Dipartimento di Ingegneria Industriale e dell'Informazione, Università degli Studi di Pavia, Via Ferrata 5, 27100 Pavia, Italy
* Correspondence: hermes.giberti@unipv.it

**Abstract:** Collaborative robots (Cobots) are compact machines programmable for a wide variety of tasks and able to ease operators' working conditions. They can be therefore adopted in small and medium enterprises, characterized by small production batches and a multitude of different and complex tasks. To develop an actual collaborative application, a suitable task design and a suitable interaction strategy between human and cobot are required. The achievement of an effective and efficient communication strategy between human and cobot is one of the milestones of collaborative approaches, which can be based on several communication technologies, possibly in a multimodal way. In this work, we focus on a cooperative assembly task. A brain–computer interface (BCI) is exploited to supply commands to the cobot, to allow the operator the possibility to switch, with the desired timing, between independent and cooperative modality of assistance. The two kinds of control can be activated based on the brain commands gathered when the operator looks at two blinking screens corresponding to different commands, so that the operator does not need to have his hands free to give command messages to the cobot, and the assembly process can be sped up. The feasibility of the proposed approach is validated by developing and testing the interaction in an assembly application. Cycle times for the same assembling task, carried out with and without the cobot support, are compared in terms of average times, variability and learning trends. The usability and effectiveness of the proposed interaction strategy are therefore evaluated, to assess the advantages of the proposed solution in an actual industrial environment.

**Keywords:** collaborative robotics; brain–computer interface; hand guiding; haptic interface; multimodal communication; assembly task





## 1. Introduction

Production in small and medium enterprises (SMEs) is usually characterized by high variability and low volumes of batches, which makes flexibility and highly adaptable production systems necessary requirements. Robotic installations easily fulfill these needs, so that their usage is quickly increasing in industry [1,2].

In SMEs, where manufacturing often relies on manual production processes, collaborative robots (cobots) are potentially easy-to-use tools that can improve productivity without loss of the advantages provided by a human-centered system [3]. Human workers indeed have an innate ability to adapt to unexpected events and to maintain strong decision-making skills, even in a dynamic and complex environment. They have flexibility and adaptability for learning new tasks and intelligence, while the robots can guarantee physical strength, repeatability, and accuracy. Human-Robot Collaboration (HRC) aims at being complementary to conventional robotics, increasing the human participation in terms of shared time and space [4]. With HRC, humans and robots can share their best skills, provided that the involved devices are designed for both safety and interaction.

A safe, close interaction with human operators is only part of the technical issues to be faced to exploit the advantages of a collaborative working station. Much effort is dedicated,

still mainly at a research level, to enhance the design of human-aware robots, able to cope with uncertainties due to human presence [5–7] and to get, at the same time, command and guidance messages from them [8]. Several communication technologies are being developed to reach a real Human-Robot Collaboration (HRC). Interaction modes, such as gesture recognition [9], vocal commands [10], haptic controls [11], and brain-computer interfaces [12] are some of the technologies exploitable to provide inputs to the robot, allowing the operators to dynamically alter the robotic behavior during its functioning or taking control of the ongoing task [8,13,14]. These technologies can also be grouped into a multimodal communication strategy, enabling the possibility of programming a robot in a more adaptive and unstructured way.

All of the above-mentioned techniques have been extensively discussed in [8], where a metric for evaluating the performances and the type of information exchanged between the human and the robot is proposed. The type of messages are classified as Command and Guidance messages, achievable through different interfaces. The former (i.e., command messages) communicates to the robot simple commands such as "next" or "stop", not requiring any kind of parameter to be exchanged. Guidance messages, on the other hand, provide instructions regarding how the robot should move, thus requiring a continuous flow of positional or force information. For this reason, command messages can be considered the simplest way of interaction, nevertheless requiring a suitable design of the programmed task.

According to [7,15], assembly tasks are typical tasks, which well suit collaborative robotic applications since the operator's action influences the behavior of the robot and vice versa. However, the most common use of collaborative robots for assembly tasks in manufacturing lines consists of workstations suited for sequential assembly [13], in which the robot performs the simpler operations and the most complex or variable ones are left to the human. Whenever possible, the worker carries out the last manipulations on the assembled product at the end of the assembly line so as to limit the need for interaction with the robot. A more complex cooperative assembly, i.e., parallel assembly, is characterized by human intervention taking place in parallel to robot activities, so that the last assembly steps can also be carried out by the robot. In this second scenario, timing and coordination between humans and robots are critical factors, which might severely affect the collaboration, which therefore strengthens the need for a suitable interaction strategy.

Different collaborative strategies can be exploited according to the degree of task interconnection and dependency [16,17]. When the possibility of a direct command message from the operator to the robot is enabled through a suitable user-interface, human's intentions can be communicated to the robot. It is then possible to alternate between Independent strategies– in which the operator and the robot work simultaneously and independently on their own tasks– and Supportive strategy– in which the operator receives assistance from the robot. The switch between Independent and Supportive phases can be enabled based on the timing given by the operator, who works independently on his own task until he gives a triggering signal to the robot [18]. This strategy can go under the name of Collaboration On-Demand since the possibility of switching from independent to supportive phase is left to the operator. Once a consent exchange is detected by the machine, the operator can be assisted by the robot in a supportive manner.

Under specific circumstances, e.g., when the presence of environmental noise or poor illumination occur, or when the operator's hands must be engaged in the assembly process and cannot be used to give commands, technologies like voice or gesture recognition might result unsuitable, and therefore the possibility to exploit techniques based on Brain–Computer Interface (BCI) is conveniently investigated. The potentiality of this techniques in applications for Industry 4.0 has been pointed out [19], but, to the authors' knowledge, few industrial applications have been actually developed. BCI is extensively studied in the neuroscience field [20–22]. Applications to robotics have been developed in different contexts [23,24], other than industrial applications.

In this work, we test the concept of collaboration on demand by setting up a Brain–Computer Interface (BCI) to transfer command messages from the operator to the cobot, thus enabling a supportive behavior in which the operator can use hand guiding control to be assisted by the robot. The communication strategy and the collaboration on-demand are deployed in a proof-of-concept assembly task developed on a TM5-700 cobot.

The BCI allows keeping the operators' hands free for the assembling task and can be adopted in noisy environments. Moreover, the possibility to insert few BCI sensors in the personal protective equipment (i.e., helmets) would pave the way for application in industrial environments.

The paper is organized as follows: Section 2 describes the overall framework of the collaboration on-demand, which adopts the BCI and the hand guiding control. In Section 3, the proof-of-concept assembly task is presented to validate the proposed solution. Results are presented in Section 4 and conclusions are drawn in Section 5.

## 2. Collaboration On-Demand Strategy Exploiting BCI-SSVEP and Hand Guiding

The different phases of an assembly process can be classified based on their repetitivity and complexity. The robot can take charge of the most repetitive and unskilled operations, while the human operator can execute complex activities. Collaborative robots in assembly tasks can also provide benefits in handling large and heavy objects [25], both thanks to hand-guided operational mode and to the capability of reducing the apparent mass of heavy work pieces by a factor of ten or more [26]. This means that the physical strain of the workers is significantly reduced.

The cooperation between human and cobot relies on the synchronization between the different phases and on a proper task assignment. In a collaboration on-demand strategy, the workcell is human-centered, and the robot is intended as a supportive tool for the worker. Figure 1 schematically represents the process by highlighting the alternating switch between two different phases, namely independent and supportive phases. During the independent phases, the robot and the operator work in a coexistence scenario, operating on different workpieces and processes. As soon as the operator needs the robot's assistance, the supportive phase can be enabled: the operator and robot work on the same workpiece interactively, and the robot must behave according to human intentions. This requires the robot to be warned of human intention in advance through a command message [8], which can be sent by a worker through a proper user-interface with the desired timing. The operator can therefore use the robot as a flexible tool whose supportive behavior is triggered at will, according to their needs and timing.

When the supportive function ends, the independent behavior can start again, and robot and operator keep on carrying out their planned activities. Furthermore, the reverse transition from supportive to independent phase can be controlled by the human through another command message so that only two different messages are needed to deploy the described interaction scheme.

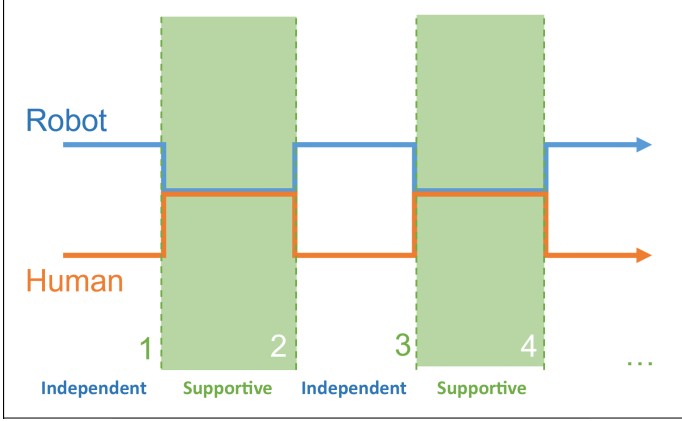

**Figure 1.** Collaboration on-demand.

In the particular industrial case proposed in this paper, the switch between independent and supportive phases is controlled through a BCI, which allows sending a command message without the use of hands. The BCI is used in a reactive mode with the Steady State Visually Evoked Potentials (SSVEP) method [19]: the operator looks at an external monitor with images blinking at two different frequencies corresponding to the two command messages needed to switch between the two operating modes. During the supportive phase, hand guiding control is exploited to position the objects in a co-manipulation mode. Hand guiding control has been realized by means of a six-component load cell mounted on the robot wrist, as described more in detail in the following paragraphs.

An exemplary assembly task has been considered as a test case consisting of pick and place operations, relative positioning of objects, and joint connections with bolts and nuts. The task can be therefore divided into simple, repetitive operations, such as pick and place, and more complex ones, such as joint connections. The former are assigned to the robot, whereas the more complex manipulations subtask can be left to the operator – as soon as the cobot has picked up the assigned component, it positions itself in a stand-by pose, waiting for the operator to take control of the process. As soon as it is ready, the operator switches to supportive operational mode so that the intermediate step of positioning the object relative to the previous one can be conducted in a cooperation mode through hand-guiding control. The robot can then assist the operator by holding and moving large and heavy objects according to its payload, leaving the operator the flexibility to properly position the component with the desired timing and then join it to the other parts of the structure being assembled.

Figure 2 shows the framework proposed, in which the operator can interact with the robot manipulator through two different interfaces. The activities related to the Brain–Computer Interface are described on the left side of the figure, whereas those related to load cell and haptic control are reported on the right side of the scheme. Looking at the left-hand side of the scheme in Figure 2, SSVEP signals are collected by electrodes, processed and then referenced to suitable Command Messages to be sent to the robot controller. On the right side of the scheme, the load cell provides a haptic interface featuring hand guiding control, which introduces Guiding Messages to properly position the robot. The task flow depends at the same time on the pre-programmed robotic subtasks and on the real-time commands given by the human operator. The latter makes the decision about when the robot must move independently and when to switch to a supportive phase.

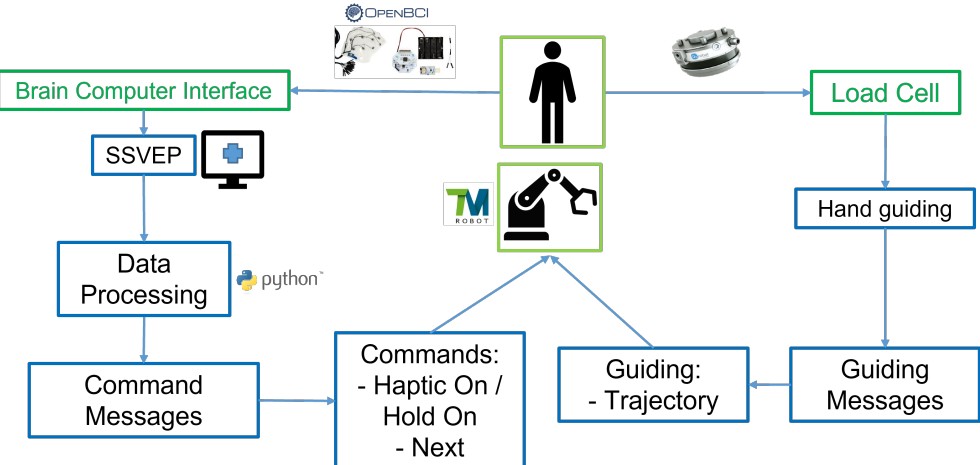

**Figure 2.** BCI-SSVEP with Load Cell sensor Framework.

Each BCI-SSVEP signal is assigned to a pre-defined specific robotic function. When the operator and the robot end their independent phases, the operator can activate the hand guiding mode by providing the command "Haptic On". The "Hold On" command is given when the operator needs the robot to keep precisely the same position in which

it has been placed through hand guiding control so that the assembly operation can take place. The same SSVEP signal can be used for the two cases. When the operator finishes connecting the two parts by screws and bolts, the "Next" command allows the robot to get back to its independent phase. The logic applied therefore requires only two signals, hence, two different stimuli: one associated with the "H" command message (which can have two different states: "Haptic On" or "Hold On"), and another for the "N" ("Next") command.

　　　Figure 3 represents the same collaboration on-demand framework from the perspective of the human operator and the robot, which are seen as parallel players in the task. Command and guidance messages are in charge of timing and controlling the task – when the operator sends the "Haptic On" command the two branches of the independent phases merge, starting the supportive phase (collaborative) [16], in which operations are headed by the operator and supported by the robot. After the hand-guided phase, in which the operator can place the component on hold by the robot relative to the structure being assembled, the "Hold On" command allows keeping the component in place so that bolting operations can take place. Afterward, the "Next" command makes the operator and robot get back to their independent phases. The former can continue to perform assembly operations for which he does not need any support, the latter can pick the next component to be handed to the operator. He will pick it up just when he needs it, and he will take charge of the process through another "Haptic On" command.

　　　The following subsections detail how BCI data are treated to manage the switching between independent and supportive collaborative phases and how the hand guiding control is deployed on the adopted Techman TM5-700 cobot.

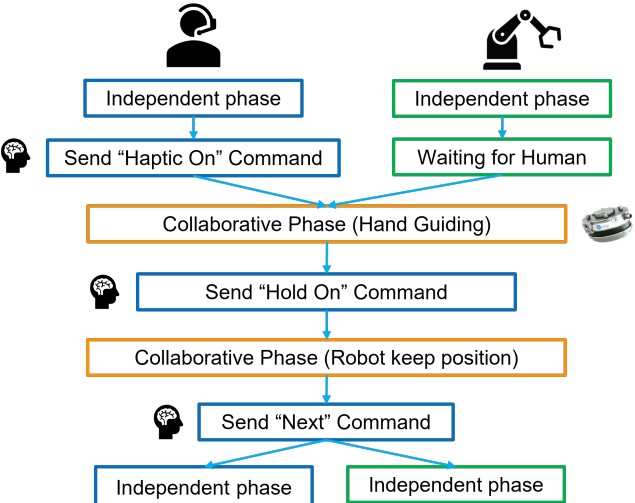

**Figure 3.** Collaboration on-demand with BCI and hand guiding.

### 2.1. Brain–Computer Interface Based on SSVEP

　　　Steady State Visually Evoked Potentials (SSVEP) response is a BCI reactive technique based on Electroencephalogram (EEG) signals, having the highest potentiality for industrial applications [19]. It allows interacting with the robot without the use of hands [27,28], gestures, or voice commands, and it does not require much training sessions. When the operator, wearing an headset, is subjected to a visual stimulus, the signals gathered from the response of the visual cortex area, suitably processed and analyzed, allow to provide inputs to the robot controller.

　　　In the proposed work, EEG signals are acquired using a cap with electrodes located in accordance with the international Ten-Twenty system [29], the representation of which is reported in Figure 4. SSVEP signals are recognized when the retina is subjected to visual stimuli blinking in the band of 3.5–75 Hz [22] even if, in common practice, the upper bound of the frequency range of the stimuli can be limited to 20 Hz [20]. SSVEP signals are characterized by a high Signal to Noise Ratio (SNR) [20,21]; hence, it is possible to

recognize different stimuli with low processing effort and good precision. The visual cortex area reflects the same frequency of the stimulus.

Two kinds of commercial electrodes can be exploited for the purpose: wet, with an appropriate electrolyte gel applied on sensors, or dry [30]. Practical use of dry electrodes would be easier in industrial environments, as it is not necessary to apply gel to allow electrical contact.

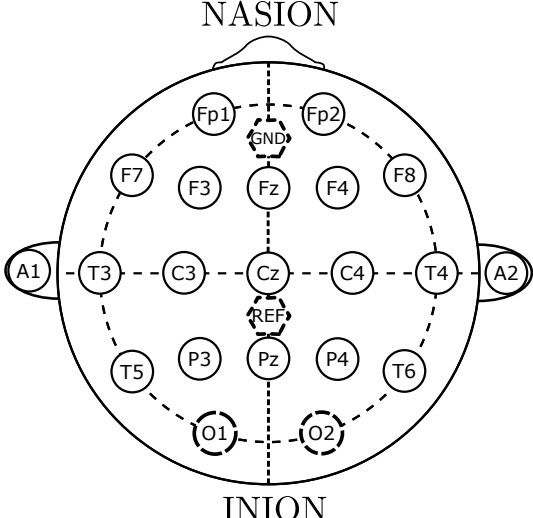

**Figure 4.** International ten-twenty system. The subset of exploited electrodes O1, O2, REF, and GND is highlighted with dashed bold line.

In order to deploy a methodology suitable for industrial applications, the lowest number of sensors has to be exploited; according to the ten-twenty system, the response of the visual cortex area can be acquired through the electrodes O1 and O2 highlighted in Figure 4. Electrodes O1 and O2 are placed on the occipital area, which is the one responsible for visual processing [31]. Ground (GND) and reference (REF) electrodes placed on the midline sagittal plane of the skull must be used for referencing and denoising the signal.

When the operator is subjected to a visual stimulus blinking at a given frequency, the same frequency and its multiples can be identified in the signal spectra. The generable visual stimuli depend on the type of devices adopted. In this work, an LCD monitor with a refresh rate of 60 Hz is used, in which two blinking windows are displayed [32,33], corresponding to the two command messages "H" and "N" needed for the robotic application. An OpenBCI headset and a Cython-Daisy board have been used in the setup, operating at a sampling rate of 125 Hz. The OpenBCI GUI software receives signals from the Cython-Daisy board over the LSL communication protocol.

The SSVEP frequency recognition software, and its integration with the robotic workflow described in the previous section, have been specifically developed. The algorithm flow is shown in Figure 5. When the robotic task starts, the first step of the SSVEP frequency recognition program is to open the communication with the OpenBCI software and flush the data buffer. Signals O1 and O2 of the visual cortex area are then gathered with a time window of 1 s. They are averaged and filtered using a digital, fourth order, Butterworth band-pass filter in the range 4–45 Hz. Finally, a Hamming window is applied to the signal, and a Fast Fourier Transform (FFT) is performed to identify the frequencies related to the visual stimulus. The FFT main peaks are analyzed to seek peaks that exceed a certain threshold at pre-set frequencies (the stimulus frequency and its first two multiples), which correspond to an actual will of the operator to give a command message.

These operations are performed in a continuous loop, and the corresponding command messages are sent to the robot only if a "cooperative flag" has been activated, meaning that the robot has completed its independent phase and is available to shift to cooperative

operational mode. This prevents false positive commands from being transmitted to the robot before its independent phase has been completed.

Robotic commands corresponding to each command message can be general subtasks programmed in the robot controller [18,23,34] or a sequence of robot movements controlled by the PC. In the present work, robotic subtasks are programmed in the robot controller and triggered by the PC when the corresponding command message is detected. The mentioned signal processing software, as well as the one in charge of sending commands to the robot, have been realized in Python 3.8 with SciPy *v1.8.1* and Pymodbus *v2.5.3* libraries.

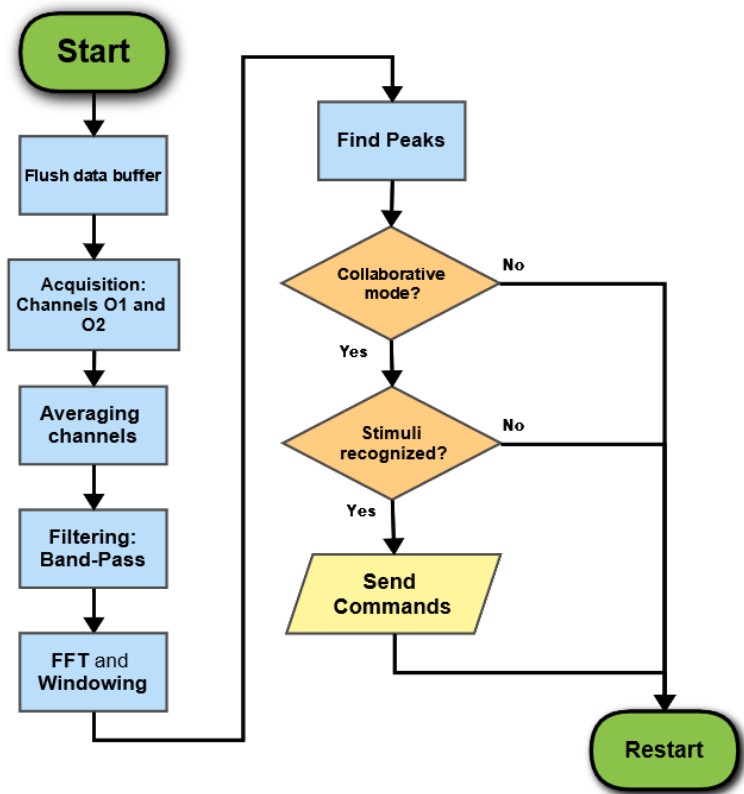

**Figure 5.** SSVEP frequency recognition software flowchart.

The issue of having a quick reactive response is essential for the fluidity of the robotic task. For this reason, to improve the system promptness, a time window length of 1 s was tested for frequency recognition in SSVEP and compared with a 2 s one [18]. The poorer spectral resolution of 1 Hz obviously affects the set of exploitable stimulus frequencies and worsens somehow the signal to noise ratio [35]. Figures 6 and 7 report the SSVEP responses to a visual stimulus blinking at 8 Hz and 10 Hz, respectively. For each frequency, the results corresponding to the time windows of 2 s and 1 s are reported in Figures 6a,7a and Figures 6b,7b , respectively. For all the considered cases, it is possible to detect the main frequency of the stimulus, with the first multiple frequencies also visible in the case of higher spectral resolution.

Since, in the present application, the operator is moving during the assembly task, the issue of motion artifacts had to be faced. Motion artifacts result in a signal spectrum with several peaks spread in the analyzed frequency range [36], which might generate false and unintended command messages. This issue had to be solved in real time during the signal processing phase by discarding all the occurrences presenting any peak at frequencies not corresponding to the expected response.

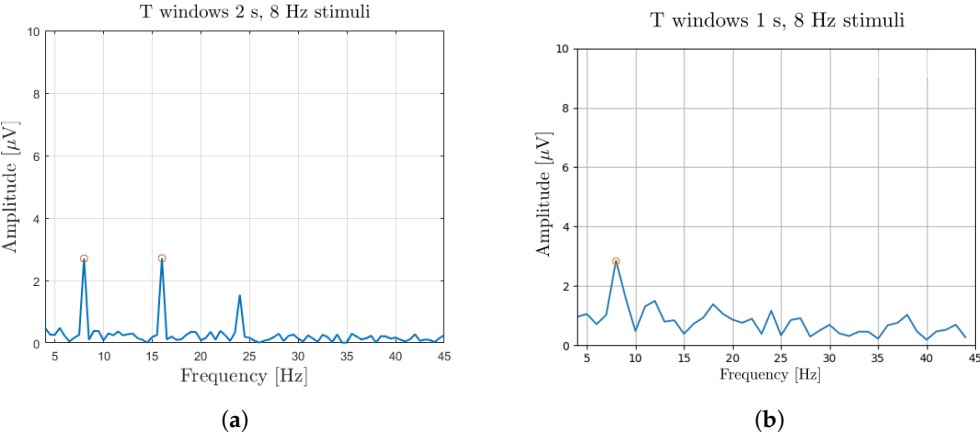

(a)

(b)

**Figure 6.** FFT analysis of SSVEP response for a 8 Hz visual stimulus. (**a**) Time window 2 s. (**b**) Time window 1 s.

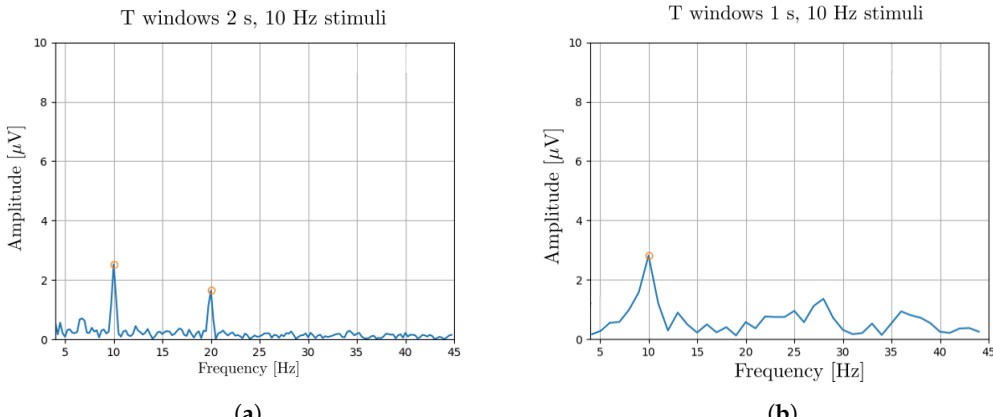

(a)

(b)

**Figure 7.** FFT analysis of SSVEP response for a 10 Hz visual stimulus. (**a**) Time window 2 s. (**b**) Time window 1 s.

### 2.2. Haptic Control Based on End Effector: Hand Guiding Control

In the proposed assembly task, positioning and co-manipulation operations require guidance messages for motion control and, in particular, for the proper positioning of the workpiece to be joined to the structure being assembled. Hand-guiding control is suitable for this purpose, assisting the operator in positioning and sustaining the weight of the parts to be assembled.

In the exemplary test bench used for demonstration, hand-guiding control has been developed by exploiting an OnRobot HEX-E v2 force sensor, measuring six axis components, mounted on the robot wrist. The system is able to sense the gripped workpiece and compensate for its weight. The arm is guided according to the forces sensed and imposed by the operator's hand, based on the proposal in [25,37,38], with a control scheme adjusted to fit the programming features of the Techman TM5-700 cobot.

A *force control* block in TMFlow software, activated during the cooperative phases, allows following any driving force provided by the operator (i.e., force in *x*, *y* and *z* directions). According to the ongoing phase in the task and, in particular, to the piece to be assembled in the pre-defined sequence, different directions for control can be enabled or disabled to have either a planar or 3D motion. For the first part to be assembled, positioning requires only a Cartesian linear motion in *x*, *y* and *z* coordinates. For the second positioning phase, the angular orientation along the Z-axis of the wrist reference frame is involved. Since only the rotation along the Z-axis is enabled, the gravity force always acts along the Z-axis of the robot's wrist.

The force control block integrates a PID control on the sensed forces and torque applied to the end-effector. To compensate for the gravity effects on the raised objects,

measurements are reset at the beginning of each force control routine, and then stored to be used as reference force/torque for the PID controller.

PID gains used for the Cartesian linear motion are: $k_p = 0.15$, $k_d = 0$, and $k_i = 0.00001$; for the rotation along the Z-axis: $k_p = 0.012$ and $k_i = k_d = 0$.

### 3. Application of the Proposed Framework in an Exemplary Assembly Task

The discussed control structure has been validated by means of an exemplary assembly task developed with a TM5-700, 6 kg payload. The same control structure can, however, be applied to a bigger robot, in which the higher admitted payload allows assembling heavier components, thus enhancing the effectiveness of the proposed solution.

The task is divided into the following steps:

1. The operator is preparing a long aluminum profile and assembling a corner joint with bolts and nuts, while the robot is picking a short aluminum profile and positioning it in front of the operator, ready to be hand-guided. During this phase, the operator and the cobot are working with independent collaboration strategies.

2. In order to activate the cooperative mode through the BCI signal ("Haptic On" command), the operator looks at the monitor window with the letter "H" blinking at a frequency $f_1 = 8$ Hz. The person can then exploit the hand-guiding control and place the beam in the final position where it is to be assembled. After a second command message is given through BCI ("Hold On" command, blinking letter H), the robot keeps the component in position, and the operator can connect the two aluminum profiles with a corner joint and nuts. In this step, all activities are carried out with supportive strategies.

3. After finishing the assembly of the two parts, the operator gives the "Next" command by looking at the "N" letter blinking on the monitor at a frequency $f_2 = 10$ Hz. The operator prepares independently a second corner joint and gathers bolts and nuts, and the cobot executes the next routine, picking another long aluminum profile, handing it to the operator, and waiting motionless for the next BCI message. In this phase, the operator and robot are working independently.

4. When ready, the operator once again activates the cooperative mode through the BCI signal ("Haptic On" command) and hand-guides the beam into the proper position to be assembled. After a second command message through BCI ("Hold On"), the robot keep the component in position, and the operator can join the two aluminum profiles with a corner joint and nuts. Once assembly is complete, the person communicate to the robot the end of the task, through the last "Next" command.

The following Figures 8–10 highlight the main steps of the assembly task: Figure 8 shows an independent phase, representative of phase numbers 1 and 3 in the numbered list above, where the operator is mounting an angular joint while the robot is bringing the beam into place. It is possible to notice in the background the monitor with the blinking windows exploited as visual stimuli for BCI.

In Figure 9, the switch from independent to supportive phase is enabled by the operator who looks at the "H" letter on the monitor, to activate the hand-guiding mode. This switch only takes 1 s, thanks to the FFT analysis discussed in the previous Section 2.1.

Finally, Figure 10 shows the supportive phase in which the operator exploits hand-guiding control to position the beam to be assembled. When a proper positioning has been achieved, the hand guiding control can be deactivated by looking at the "H" window again, so that the robot holds the component steady in the desired position. Then, when the operator finishes the assembly, the robot can be moved by looking to the "N" window. In this way, the proper task timing is given by the operator, so that the robotic code is able to obviate any difficulties or setbacks in the assembly phase.

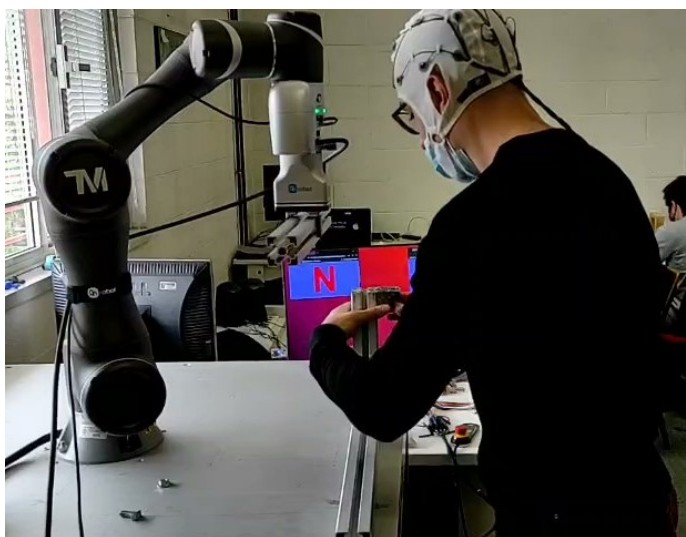

**Figure 8.** Independent phase. Step 3 of the assembly task.

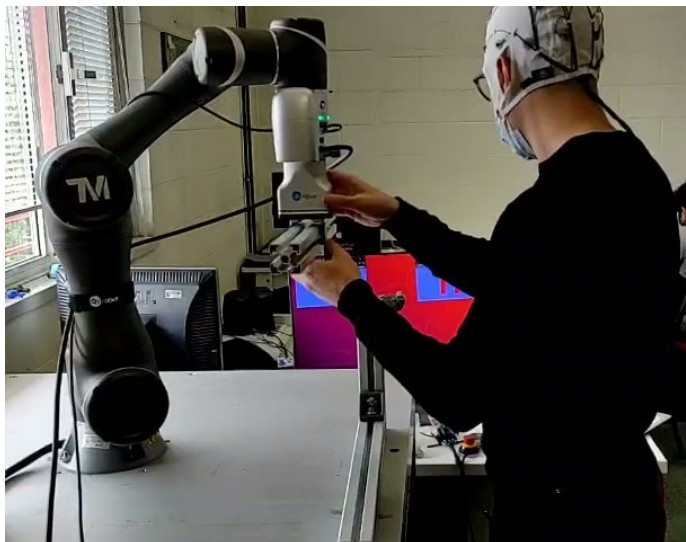

**Figure 9.** The operator looking at the "H" window is activating the hand guiding.

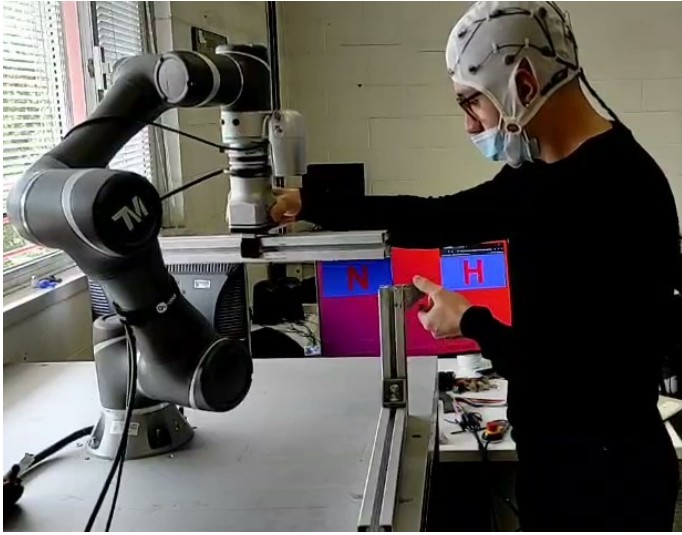

**Figure 10.** Step 4 of the assembly task. Supportive phase with hand guiding.

Figure 11 shows the flowchart of the robotic program. Independent phases are represented with yellow blocks (continuous line contour), decision phases in which the robot is waiting for BCI inputs in blue (rhomboidal shape, dashed line contour), and supportive phases in green (dashed line contour).

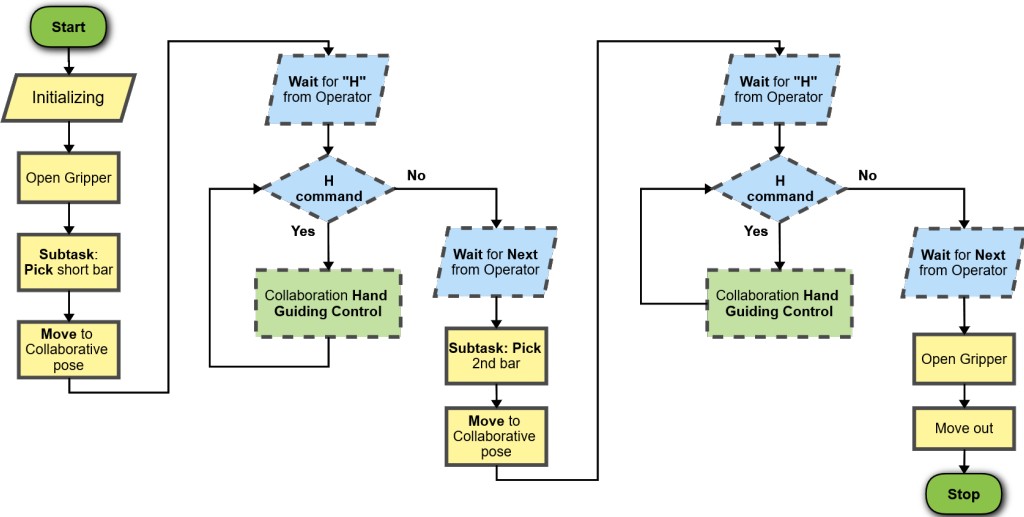

**Figure 11.** The robotic task flowchart.

Right after the start block, initialization actions and variable setting take place. The subtasks to be executed in independent mode are then carried out (i.e., Open Gripper, Pick bar Subtask, Move to handing pose where to wait for the operator). When the independent phase is over, the cobot is ready for entering the cooperative mode, it activates the "cooperative flag" enabling the possibility to receive the BCI command "H" from the operator. Then it waits for the operator's command: measurements from the BCI device and results from data analysis start being taken into account by the robot controller. SSVEP frequencies gathered from the BCI headset are compared with the pre-set frequencies to recognize the operator's choice. The program makes a step forward when the frequency corresponding to the H signal is detected.

The H command can have two binary statuses (i.e., 1, yes; 0, no). In the former status, the hand-guiding control is activated. In the latter, the hand-guiding is deactivated and the system goes into a new decisional block, waiting for the BCI next (N) command. At this point, the second independent subtask starts and the described cycle is repeated.

## 4. Results

In order to assess the potential improvements achievable through the described collaborative strategy, two series of tests have been carried out by repeating the assembly twenty times firstly purely manually and then with robotic assistance. Both assembly series were repeated by two different operators (named in the following S and Y), to also take into account variability due to the human factor. The cycle times were measured for each repetition.

Figure 12a,b show the distributions of the cycle times corresponding to the two operators, S and Y, respectively. In each figure, the cycle times related to the manual assembly (labeled with *Not assisted*), and to the human/robotic assembly (labeled with *Assisted*) are reported, together with the values of average $\mu$ and standard deviation $\sigma$ and the representation of the corresponding Gaussian distribution.

Both S and Y operators experienced a clear reduction of the average cycle times in the assisted case compared to the not assisted one (71.3 s against 101.05 s for operator S and 86.25 s against 102.3 s for operator Y, corresponding to a reduction of 29.44% and 15.69% respectively), mainly thanks to the assistance of the robot in properly positioning the component to be assembled by means of the hand guiding control and holding it in the

proper position during the bolting. Moreover, the cases in which the operator is assisted by the cobot show a lower standard deviation of cycle times, meaning that the presence of a predetermined workflow mitigates the variability of the cycle times, helping the operator be more regular in operations.

The standard deviation of the cycle times is equal to 6.19 s against 10.47 s for operator S, and 8.28 s against 12.49 s for operator Y.

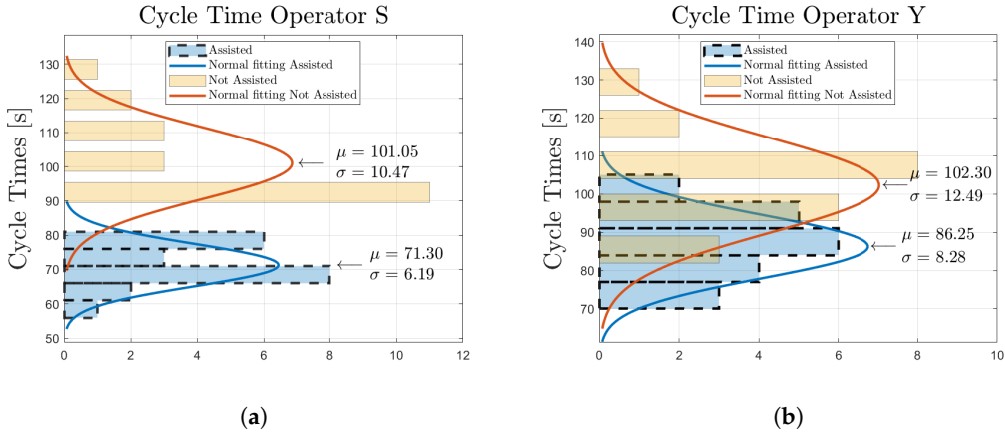

**Figure 12.** Assembly cycle times distributions for Assisted and Not Assisted cases with mean and standard deviation. (**a**) Operator S. (**b**) Operator Y.

Figure 13 shows the same cycle times as in Figure 12, reported this time in sequential order, as gathered during the execution of the test series. A learning effect is visible in all the series of tests, with a reduction of the cycle time occurring when the number of repetitions of the task increases, due to the fact that the operators improve their skills during the activity. The linear regression lines plotted in the figures can be used to estimate the learning rate of each operator. The linear regression slopes in the assisted cases show a reduction of the cycle time for both the operators (0.78 s/iteration for operator S, 0.51 s/iteration for operator Y), highlighting a beneficial effect of the robotic system. On the other hand, in the not assisted case, a more erratic behavior is observed, with operator S showing almost no learning (0.15 s/iteration), and operator Y showing a relevant learning rate (0.74 s/iteration). The results are summarized in Table 1.

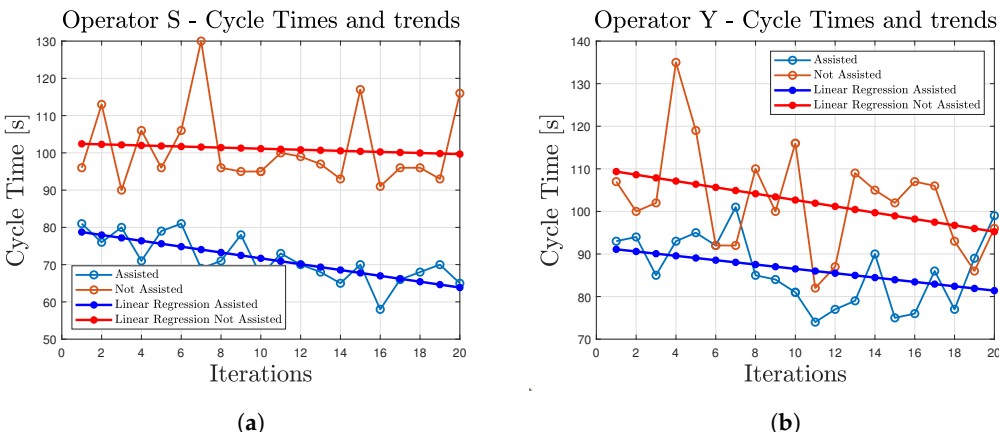

**Figure 13.** Learning rates. (**a**) Operator S. (**b**) Operator Y.

**Table 1.** Results summary table: Assisted vs Not Assisted.

| Assisted vs. Not Assisted Cases Summary Table | | | |
|---|---|---|---|
| **Operator** | **Avg. Cycle Time $\mu$** | **Cycle Time $\sigma_{STD}$** | **Learning Rate** |
| Operator S — Not assisted | 101.05 s | 10.47 s$^2$ | 0.15 s/iteration |
| Operator S — Assisted | 71.30 s | 6.19 s$^2$ | 0.78 s/iteration |
| Operator S — Percentage difference | −29.44% | −40.88% | 80.77% |
| Operator Y — Not assisted | 102.30 s | 12.49 s$^2$ | 0.74 s/iteration |
| Operator Y — Assisted | 86.25 s | 8.28 s$^2$ | 0.51 s/iteration |
| Operator Y — Percentage difference | −15.69% | −33.71% | −31.08% |

The following paragraph points out all possible issues related to wrong or poor signals from the interfaces which might arise during the development of the task.

Regarding command messages (i.e., through BCI), the following cases can be highlighted:

- A low signal-to-noise ratio, caused by a poor quality of the electrode/gel and skin contact, can cause a non-acquisition of the frequencies associated with command messages, leading to a slowdown of the task, as the robot will keep on waiting for a command through BCI.
- In the presence of motion artefacts, the commands are ignored. This is a positive feature if the motion artefact is to be discarded, but, if motion artefacts occur right when the operator is turning their head to look at the blinking signal, the desired command message would not be delivered. A new signal acquisition would be required, causing a delay of a few seconds in the overall task.
- Peripheral vision of operators might lead to the acquisition of incorrect command during the task, even if the operator is not purposely looking at the blinking signals. The influence of peripheral vision can be reduced by locating the blinking stimuli far from the peripheral field of view.

As for the guidance messages (i.e., through hand-guiding control):

- A wrong guidance message can be generated only if the operator guides the robot in a singularity position. Since the hand guiding control uses the motion control operating in the working space, in singularity positions, the motion control fails to compute the inverse kinematic, leading to a failure of the collaborative mode. This possibility can be completely avoided, and can be considered as human error.

## 5. Conclusions

A collaborative on-demand strategy is implemented in the present work, with the aim of demonstrating the possibility of exploiting the Brain–Computer Interface to give command messages to the robot controller in an industrial assembly task. The BCI provides the operator with the chance for controlling and giving proper timing to the robotic task during the assembly operations, without the need to use hands to push physical buttons or interact with a gesture recognition system.

A proper BCI command can switch the robot operational mode from independent to cooperative, thus activating a hand-guiding control by which the cobot and the human operator can interact in a supportive manner during the assembly task. The performance of this approach has been experimentally validated with two different operators, resulting in a significant reduction not only of the average cycle time (−29.44% and −15.69%), but also of its variability (i.e., standard deviation of the cycle times), thus leading to a more predictable productivity.

Nowadays, the BCI technology is not commonly applied in robotics for industrial applications, for example in assembly tasks. The proposed work demonstrates that if BCI electrodes could be fitted in the helmets provided as personal protective equipment in an industrial environment, this technology could be a considerable option for Industry 4.0 applications.

However, some limitations are still present. The human involvement in the completion of the shared task requires the overall assembly process to be clearly structured in advance,

having clear in mind the alternative phases to be assigned to the human operator and to the robot. This introduces strict constraints in the design phase of the task.

Compared to standard robotic applications, the fact that the timing control of the assembly task is entrusted to the human operator may introduce delays due to the operator's behavior. Some might be due to human errors, but others are related to technological limitations of the selected interface: the signal to noise ratio in data generated by BCI, normally rather high, might show relevant variations depending on the subject wearing the BCI helmet. In such circumstances, the peak of the expected frequency may require more than one time-window to be detected, negatively affecting the overall execution time.

Motion artifacts, faced in this work by discarding all the occurrences presenting any peak at frequencies not corresponding to the expected response, might also occur when the operator looks at the blinking signal to generate the command message. In such a circumstance, the corresponding command would not be given, implying a delay due to the need of a new signal acquisition.

For the explained reason, the issue of signal-to-noise ratio, depending on the subject wearing the helmet, and motion artifact are still issues to be more comprehensively investigated in a further development of the work.

**Author Contributions:** Conceptualization, H.G. and M.C.; methodology, M.C., Y.D. and F.I.; software, Y.D. and F.I.; validation, M.C., F.I. and Y.D.; formal analysis, H.G. and M.C.; investigation, M.C., F.I. and Y.D.; resources, H.G.; data curation, M.C. and F.I.; writing—original draft preparation, F.I.; writing—review and editing, M.C.; visualization, Y.D., F.I. and M.C.; supervision, H.G. and M.C. All authors have read and agreed to the published version of the manuscript.

**Funding:** This research received no external funding.

**Institutional Review Board Statement:** Ethical review and approval were waived for this study, since the manuscript only involves frequency data in reaction to a blinking screen frequency.

**Informed Consent Statement:** Informed consent was obtained from all subjects involved in the study. Written informed consent has been obtained from the involved subjects to publish this paper.

**Data Availability Statement:** The data presented in this study are available on request from the corresponding author.

**Acknowledgments:** The authors would like to thank Stefano Ciardiello for carrying out part of the experimental tests during his BSc thesis.

**Conflicts of Interest:** The authors declare no conflict of interest.

## Abbreviations

The following abbreviations are used in this manuscript:

| | |
|---|---|
| BCI | Brain–Computer Interface |
| SSVEP | Steady State Visually Evoked Potentials |
| SNR | Signal to Noise Ratio |
| EEG | Electroencephalography |
| HRC | Human Robot Collaboration |
| FFT | Fast Fourier Transform |

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
