# Peer review of "Brain–Computer Interface and Hand-Guiding Control in a Human–Robot Collaborative Assembly Task"

_machines, doi:10.3390/machines10080654_

Round 1
Reviewer 1 Report
- SSVEP method, what do the abbreviations mean?
- Authors could further explore the concept of human-robot collaboration
- Authors say: "..... Therefore, the possibility of exploiting BCI techniques...", could you explain more, because BCI is a technique, what are those BCI techniques?
- In the framework proposed, it is not clear the collaborative phase. In addition, authors must explain because O1 and O2 were selected, supported by references.
- What are the limits?
Author Response
Reviewer 1
SSVEP method, what do the abbreviations mean?
The extended meaning of SSVEP acronym, Steady State Visually Evoked Potentials, has been added in line 135 of the paper.
Authors could further explore the concept of human-robot collaboration
To provide a clear explanation of the Human-Robot Collaboration the following sentences have been added in the introduction at LINE 33: “Human-Robot Collaboration (HRC) aims at being complementary to conventional robotics, increasing the human participation in terms of shared time and space [3]. With HRC, humans and robots can share their best skills, provided that the involved devices are designed for both safety and interaction.”
Moreover, the following reference has been added:
[3] Vicentini, F. (October 12, 2020). "Collaborative Robotics: A Survey." ASME. J. Mech. Des. April 2021; 143(4): 040802. https://doi.org/10.1115/1.4046238
- Authors say: "..... Therefore, the possibility of exploiting BCI techniques...", could you explain more, because BCI is a technique, what are those BCI techniques?
We agree with the reviewer's comment. BCI is rather an interface, and there are several techniques to process its signals. Some examples, in addition to the SSVEP response analysis exploited in this work, are the “motor imagery” active BCI technique, the “P300” reactive BCI techniques, the “cognitive load monitoring” passive BCI technique. The cited paper [17] carried out a detailed literature review to investigate the main challenges and present criteria relevant to the deployment of BCI applications for Industry.
[17] Douibi, K.; Le Bars, S.; Lemontey, A.; Nag, L.; Balp, R.; Breda, G. Toward EEG-Based BCI Applications for Industry 4.0: Challenges and Possible Applications. Frontiers in Human Neuroscience 2021, 15. doi:10.3389/fnhum.2021.705064.
The sentence at LINE 86 has been modified as follows: “and therefore the possibility to exploit techniques based on Brain Computer Interface (BCI) is conveniently investigated”
- In the framework proposed, it is not clear the collaborative phase. In addition, authors must explain because O1 and O2 were selected, supported by references.
With HRC, human operators and robots share their skills, working space and time, in order to complete the collaborative task. In the paper, at LINE 184, it is better pointed out that the supportive phase is a way of the robot to collaborate to the task completion. The concept is supported with reference to [15], which has been added at LINE187.
The term “supportive phase” has been modified to “collaborative supportive phase” at LINES 184 and 194, where we tried to point out with the description of the framework in Figure 3.
LINE 218: the following sentence has been added: “Electrodes O1 and O2 are placed on the occipital area which is the responsible for the visual processing [29]”.
To further substantiate the choice, the following reference has been added:
[29] Xing, X., Wang, Y., Pei, W. et al. A High-Speed SSVEP-Based BCI Using Dry EEG Electrodes. Sci Rep 8, 14708 (2018). https://doi.org/10.1038/s41598-018-32283-8
- What are the limits?
We tried to highlight some limitations of the proposed work in the conclusions. The following sentences have been added to the paper at LINE 418:
“However, some limitations are still present. The human involvement in the completion of the shared task requires the overall assembly process to be clearly structured in advance, having clear in mind the alternative phases to be assigned to the human operator and to the robot. This introduces strict constraints in the design phase of the task.
Compared to standard robotic applications, the fact that the timing control of the assembly task is entrusted to the human operator may introduce delays due to the operator's behaviour. Some might be due to human errors, but others are related to technological limitations of the selected interface: the signal-to-noise ratio in data generated by BCI, normally rather high, might show relevant variations depending on the subject wearing the BCI helmet. In such circumstances, the peak of the expected frequency may require more than one time- window to be detected, negatively affecting the overall execution time.
Motion artifacts, faced in this work by discarding all the occurrences presenting any peak at frequencies not corresponding to the expected response, might also occur when the operator looks at the blinking signal to generate the command message. In such a circumstance the corresponding command would not be given, implying a delay due to the need of a new signal acquisition.
For the explained reason, the issue of signal-to-noise ratio, depending on the subject wearing the helmet, and motion artifact are still issues to be more comprehensively faced in a further development of the work.”

Reviewer 2 Report
Authors research targets a very interesting issue: the human robot collaboration. Though the paper is very well written I would like to suggest some points for improvement:
1) Please provide the explanations to the abbreviations used throughout the text the first time that they appear. Although all abbreviations are explained this is not happening at their first appearance all the times.
2) In the application section please define also if the cobots can be tested for their accuracy (if applicable) and how these tests can be performed
3) In the results session please define the possible errors the cobots may do or even the possibility of not performing at all a command
4) In the conclusions part it would be nice to see future steps (or tests) of this research.
5) Please correct in Figure 12a the order of appearance of μ and σ to be similar as in the other figures.
Author Response
Reviewer 2
Authors research targets a very interesting issue: the human robot collaboration. Though the paper is very well written I would like to suggest some points for improvement:
1) Please provide the explanations to the abbreviations used throughout the text the first time that they appear. Although all abbreviations are explained this is not happening at their first appearance all the times.
The extended meaning of SSVEP acronym, Steady State Visually Evoked Potentials, has been added in line 136 of the paper. The authors verified that each first occurrence of the abbreviation is fully explained.
2) In the application section please define also if the cobots can be tested for their accuracy (if applicable) and how these tests can be performed
In the present work, reference was made to the manufacturer datasheet of the TM5-700 cobot adopted (payload 6kg, the range 700 mm). The declared accuracy is equal to 0.05mm.
In the proposed application, the cobot basically has the duty of delivering the component to the operator who take it and then positions it. For this reason, the cobot accuracy does not affect the performance of the assembly task and a further investigation was not considered necessary.
3) In the results session please define the possible errors the cobots may do or even the possibility of not performing at all a command
The following paragraph has been added to the result section (LINE 399), defining the possibility of delaying the execution of a command or the possible errors that may arise.
“The following paragraph points out all possible issues related to wrong or poor signals from the interfaces which might arise during the development of the task.
Regarding command messages (i.e. through BCI), the following cases can be highlighted:
- A low signal-to-noise ratio, caused by a poor quality of the electrode/gel and skin contact, can cause a non-acquisition of the frequencies associated with command messages, leading to a slowdown of the task as the robot will keep on waiting for a command through BCI.
- In the presence of motion artefacts, the commands are ignored. This is a positive feature if the motion artefact is to be discarded, but, if motion artefacts occur right when the operator is turning his head to look at the blinking signal, the desired command message would not be delivered. A new signal acquisition would be required, causing a delay of few seconds in the overall task.
- Peripheral vision of operators might lead to the acquisition of incorrect command during the task, even if the operator is not purposely looking at the blinking signals. The influence of peripheral vision can be reduced by locating the blinking stimuli far from the peripheral field of view.
As for the guidance messages (i.e. through hand guiding control):
- a wrong guidance message can be generated only if the operator guides the robot in a singularity position. Since the hand guiding control uses the motion control operating in the working space, in singularity positions the motion control fails to compute the inverse kinematic, leading to a failure of the collaborative mode. This
possibility can be completely avoided, and can be considered as human error.
4) In the conclusions part it would be nice to see future steps (or tests) of this research.
After discussing the limitations of the proposed application and the used technologies, the future investigation fields are reported. The following text has been added at the LINE418:
“However, some limitations are still present. The human involvement in the completion of the shared task requires the overall assembly process to be clearly structured in advance, having clear in mind the alternative phases to be assigned to the human operator and to the robot. This introduces strict constraints in the design phase of the task.
Compared to standard robotic applications, the fact that the timing control of the assembly task is entrusted to the human operator may introduce delays due to the operator's behaviour. Some might be due to human errors, but others are related to technological limitations of the selected interface: the signal-to-noise ratio in data generated by BCI, normally rather high, might show relevant variations depending on the subject wearing the BCI helmet. In such circumstances, the peak of the expected frequency may require more than one time- window to be detected, negatively affecting the overall execution time.
Motion artifacts, faced in this work by discarding all the occurrences presenting any peak at frequencies not corresponding to the expected response, might also occur when the operator looks at the blinking signal to generate the command message. In such a circumstance the corresponding command would not be given, implying a delay due to the need of a new signal acquisition.
For the explained reason, the issue of signal-to-noise ratio, depending on the subject wearing the helmet, and motion artifact are still issues to be more comprehensively faced in a further development of the work.”
5) Please correct in Figure 12a the order of appearance of μ and σ to be similar as in the other figures.
The figure has been modified.

Reviewer 3 Report
The report presented is written in great detail and in a high style. Many figures and photographic material are presented. Over 40 sources were used.
- The presented scientific report lacks:
- Equations;
- Tables.
I recommend adding equations (mathematical and other dependencies and relationships, etc.) and comparison tables (eg Figures 6, 7, 12 and 13 and/or others) if possible. Tables are especially important for visualizing and understanding the details of the data for a wide range of stakeholders. This will enhance work at the scientific level.
- The Acknowledgments field is left blank. It is desirable to add information. It is important to have a direct relationship of the contributions in the publication to the stakeholders.
- I recommend clarifying the health and ethical features specific to scientific research.
- The conclusion can be extended.
Author Response
Reviewer 3
The report presented is written in great detail and in a high style. Many figures and photographic material are presented. Over 40 sources were used.
- The presented scientific report lacks:
- Equations;
- Tables.
I recommend adding equations (mathematical and other dependencies and relationships, etc.) and comparison tables (eg Figures 6, 7, 12 and 13 and/or others) if possible. Tables are especially important for visualizing and understanding the details of the data for a wide range of stakeholders. This will enhance work at the scientific level.
Table 1, making a synthesis and comparison of the results, has been added to the paper in the results section.
- The Acknowledgments field is left blank. It is desirable to add information. It is important to have a direct relationship of the contributions in the publication to the stakeholders.
The research did not receive any external funding.
The following Acknowledgment has been added: “The authors would like to thank Mr. Stefano Ciardiello for carrying out part of the experimental tests during his BSc thesis.”
- I recommend clarifying the health and ethical features specific to scientific research.
Since the manuscript only involves frequency data in reaction to a blinking screen frequency, we did not consider any issues related to health and we did not consider involving the Ethics Committee. This was preliminarily discussed with the editor’s agreement.
- The conclusion can be extended.
The conclusion has been extended by adding limitations and possible future development. The following sentences have been added at the LINE418 in the conclusion:
“However, some limitations are still present. The human involvement in the completion of the shared task requires the overall assembly process to be clearly structured in advance, having clear in mind the alternative phases to be assigned to the human operator and to the robot. This introduces strict constraints in the design phase of the task.
Compared to standard robotic applications, the fact that the timing control of the assembly task is entrusted to the human operator may introduce delays due to the operator's behaviour. Some might be due to human errors, but others are related to technological limitations of the selected interface: the signal-to-noise ratio in data generated by BCI, normally rather high, might show relevant variations depending on the subject wearing the BCI helmet. In such circumstances, the peak of the expected frequency may require more than one time- window to be detected, negatively affecting the overall execution time.
Motion artifacts, faced in this work by discarding all the occurrences presenting any peak at frequencies not corresponding to the expected response, might also occur when the operator looks at the blinking signal to generate the command message. In such a circumstance the corresponding command would not be given, implying a delay due to the need of a new signal acquisition.
For the explained reason, the issue of signal-to-noise ratio, depending on the subject wearing the helmet, and motion artifact are still issues to be more comprehensively faced in a further development of the work.”
